# Identification of Candidate Genes Regulating Drought Tolerance in Pearl Millet

**DOI:** 10.3390/ijms23136907

**Published:** 2022-06-21

**Authors:** Animikha Chakraborty, Aswini Viswanath, Renuka Malipatil, Janani Semalaiyappan, Priya Shah, Swarna Ronanki, Abhishek Rathore, Sumer Pal Singh, Mahalingam Govindaraj, Vilas A. Tonapi, Nepolean Thirunavukkarasu

**Affiliations:** 1ICAR-Indian Institute of Millets Research, Hyderabad 500030, India; chakrabortyanimikha@gmail.com (A.C.); aswiniashaviswanath@gmail.com (A.V.); malipatilrenuka@gmail.com (R.M.); sreejanani31@gmail.com (J.S.); shahpriya1422@gmail.com (P.S.); swarna@millets.res.in (S.R.); tonapi@millets.res.in (V.A.T.); 2International Crops Research Institute for the Semi-Arid Tropics (ICRISAT), Patancheru 502324, India; abhishek.rathore@cgiar.org; 3ICAR-Indian Agricultural Research Institute, New Delhi 110012, India; sumerpalsingh@yahoo.com

**Keywords:** pearl millet, drought, functional genes, candidate genes, molecular mechanisms

## Abstract

Pearl millet is an important crop of the arid and semi-arid ecologies to sustain food and fodder production. The greater tolerance to drought stress attracts us to examine its cellular and molecular mechanisms via functional genomics approaches to augment the grain yield. Here, we studied the drought response of 48 inbreds representing four different maturity groups at the flowering stage. A set of 74 drought-responsive genes were separated into five major phylogenic groups belonging to eight functional groups, namely ABA signaling, hormone signaling, ion and osmotic homeostasis, TF-mediated regulation, molecular adaptation, signal transduction, physiological adaptation, detoxification, which were comprehensively studied. Among the conserved motifs of the drought-responsive genes, the protein kinases and *MYB* domain proteins were the most conserved ones. Comparative in-silico analysis of the drought genes across millet crops showed foxtail millet had most orthologs with pearl millet. Of 698 haplotypes identified across millet crops, MyC2 and Myb4 had maximum haplotypes. The protein–protein interaction network identified *ABI2*, *P5CS*, *CDPK*, *DREB*, *MYB*, and *CYP707A3* as major hub genes. The expression assay showed the presence of common as well as unique drought-responsive genes across maturity groups. Drought tolerant genotypes in respective maturity groups were identified from the expression pattern of genes. Among several gene families, ABA signaling, TFs, and signaling proteins were the prospective contributors to drought tolerance across maturity groups. The functionally validated genes could be used as promising candidates in backcross breeding, genomic selection, and gene-editing schemes in pearl millet and other millet crops to increase the yield in drought-prone arid and semi-arid ecologies.

## 1. Introduction

Pearl millet [*Pennisetum glaucum* (L.) R.Br.] is an important small-grained cereal crop that belongs to the family Poaceae and subfamily Panicoideae. Pearl millet is a highly cross-pollinated diploid (2n = 14) species with a genome size of ~1.79 Gb [1]. It is widely grown in arid and semi-arid regions of Africa and Asia, which are characterized as zones of low rainfall and high temperature. In India, pearl millet is the third-most cultivated food crop grown in seven million hectares of land with a productivity of 8.6 million tons in the year 2018–19 [2]. The presence of a high level of iron and zinc in the grain is important to combat micronutrient malnutrition. Drought is a major environmental stress factor across production ecologies that severely limits plant growth and development and often results in lowering crop yield. Although pearl millet is a hardy crop, drought during critical stages is the major production constraint in the semi-arid and arid regions. Thus, examining its cellular and molecular mechanisms of stress tolerance via functional genomics approaches is very important to develop better cultivars in pearl millet and sustain food and nutritional security.

To overcome the drought condition, plants have evolved a set of complex interacting layers of morphological, physiological, and biochemical responses. Morphological traits including root architecture, stay green property, leaf rolling, and yield factors are important for imparting drought tolerance and are mainly meant for efficient water absorption from the soil and conservation in the system. Root diameters and root-length density are some of the known root traits which contribute to productivity under drought conditions [3]. Pearl millet root system is characterized by fast primary root development at the early growth stage and quickly colonizing deeper soil horizon, which enables procuring water from the deeper root zone [4]. Stay green is a characteristic of some plants to extend the duration of photosynthesis by delaying their leaves’ senescence for a longer period. Stay green properties were observed in pearl millet when exposed to the drought stress, where the tolerant lines exhibited more water extraction after anthesis rather than before anthesis [5]. Leaf-rolling and wilting are some means to reduce water loss by reducing the surface area of the leaf thereby decreasing the rate of transpiration [6]. In rice, over-expression of *Abaxially Curled Leaf 1* (*ACL 1*) and *ACL* 2 regulate the leaf rolling under water-stress conditions) [7].

Physiological parameters of drought tolerance include stomatal closure, decreased photosynthesis, water-use efficiency (WUE), transpiration efficiency, and accumulation of osmoprotectants such as proline, glycine betaine, and [8]. Under drought stress, the major strategy by which plants overcome water loss is through stomatal closure. In Arabidopsis, the *GTL1* controls several stomatal traits, by regulating the rate of transpiration, stomatal density, and WUE [9]. Photosynthesis is another physiological event severely affected by water stress. Genes encoding galactinol synthase (*AtGolS1* and *AtGolS2*) showing tolerance to drought via accumulation of endogenous galactinol and raffinose were identified in Arabidopsis [10]. Under drought stress, the pearl millet genotypes accumulate organic solutes such as glucose, sucrose, amino acids, and proline which act as osmo protectants, ultimately contributing to drought tolerance [11]. Drought-exposed sorghum plants showed a low transpiration rate and slow water absorption from the soil which in turn increased the transpiration efficiency of the crop [12].

Biochemical traits in drought tolerance include reduced accumulation of reactive oxygen species (ROS) through the production of antioxidants, which act enzymatically and non-enzymatically scavenging the ROS [13]. Enzymatic component of ROS scavenging includes ascorbate peroxidase (APX), SOD, CAT, and NADH, and non-enzymatic antioxidants include proline, carotenoids, flavonoids, ascorbic acid, reduced glutathione (GSH) [14]. Drought stress enhances the glutathione reductase activity in the roots and leaves, which facilitates the defense against ROS [15]. Over-expression of *PcAPX* (ascorbate peroxidase) from *Populus tomentosa* enhanced tolerance to drought in tobacco plants by acting as ROS scavenger [16]. Several hormones are involved in stress-mediated signaling pathways, in which ABA acts as a key regulator, responsible for regulating the expression of stress-associated genes. *SnRK2* family gene *SnRK2.6/OST1* acts as a positive regulator, inducing ABA-mediated stomatal closure [17]. In cotton, under drought conditions, the over-expression of ABA-induced genes *AREB1/2* and *GhCBF3* resulted in higher chlorophyll, proline, and relative water contents [18].

At the molecular level, the response of plants to the drought condition is a multi-genic trait induced by both regulatory and functional genes [19]. Several genes and proteins having functional roles in conferring stress tolerance were identified in many crops [20,21]. These stress-associated genes are responsible for synthesizing various regulatory proteins, detoxifying enzymes, proline, and other compatible [22]. In pearl millet, PgGPx encodes a protein, which acts as a ROS scavenger during drought stress [23]. Heat shock proteins (HSP) acting as molecular chaperones, were played role in protein unfolding and denaturation control [24]. The heat shock proteins involved in heat-responsive pathways directly correlated to high-temperature tolerance [25]. Late embryogenesis abundant (LEA) proteins, aquaporins (AQP), and membrane-stabilizing proteins have also been demonstrated, with crucial roles in increasing water-binding capacity under drought stress. Additionally, several TFs involved in modulating gene expression under drought conditions, including members of MYB, MYC, DREB/CBF, ABF/AREB, NAC, bZIP, and WRKY families, were studied in maize [20,21], and bean [26].

Pearl millet is cultivated in a variety of production ecologies which differs in soil, weather, and resource factors. Drought at critical stages, namely the seedling, vegetative, flowering, and grain-filling stages, costs grain and fodder yield, especially in arid and hyper-arid ecologies. Though drought stress causes a serious impact at every critical stage, a reduction in fertile florets at the flowering stage leads into low productivity. Genes responsible for the adaptation to the drought stress especially during the critical flowering stage can provide a much-needed boost to the grain and fodder yield. Hence, an experiment was conceived to understand the genes involved in regulating drought tolerance in pearl millet inbreds belonging to four well-known maturity groups at the flowering stage; to structurally and functionally characterize drought-responsive genes representing eight functional groups; to understand the orthologs and haplotype patterns of the genes across pearl millet, sorghum, fox millet, proso millet, and finger millet; to identify the drought-tolerant genotypes based on the differential expression pattern; and to find out the utility of these genes in an applied breeding program to develop tolerant varieties.

## 2. Materials and Methods

### 2.1. Identification of Drought Gene Homologs and Chromosomal Location

A set of 171 genes were identified from the drought database (DroughtDB) [27] belonging to different functional and regulatory characteristics related to drought. The nucleotide sequences of the selected genes were retrieved from NCBI (https://www.ncbi.nlm.nih.gov/; (accessed on 15 December 2020)), TAIR (https://www.arabidopsis.org/index.jsp; (accessed on 15 December 2020)), maizeDB (https://www.maizegdb.org/), RGAP (http://rice.plantbiology.msu.edu/index.shtml; (accessed on 15 December 2020)) databases and were searched against *Cenchrus americanus* (pearl millet) (ref: ASM217483v2) genome to find out the homologs. The nucleotide sequences were blasted using different parameters to find the homology (percentage identity more than 85%, e-value less than 1 × 10^−5^, and a minimum length of match 150). The identified drought genes were mapped to the pearl millet chromosomes based on the location. AUGUSTUS (https://bioinf.uni-greifswald.de/augustus/; (accessed on 25 May 2022)) was used to obtain the drought gene structures—exons, introns, and CDS based on the alignments of their coding sequences [28].

### 2.2. Plant Material and Induction of Drought Stress

A set of 48 genetically diverse pearl millet inbreds representing four maturity groups as well as corresponding to four distinct production ecologies (very-early, early, medium, and late maturity groups) was used as experimental materials for this study (Appendix A). These genotypes were selected based on the initial screening of a large set of pearl millet inbreds under drought. The 48 pearl millet genotypes received from ICRISAT, India belonged to four maturity groups based on the 50% flowering time, namely nine from very early (less than 45 days), nine from early (45 to 50 days), 13 from medium (50 to 55 days) and 17 from late (more than 55 days). Around 4 to 5 seeds of each genotype were sown in six pots where three pots each, also considered three replicates, were designated as control and drought. The pots were filled with red-loamy soil and were watered regularly till the 50% flowering stage. At the time of 50% flowering, drought stress was induced in the drought-designated pots by withholding water for 15 consecutive days according to the flowering stage of the genotypes, while control plants were watered regularly.

### 2.3. RNA Isolation

For the expression study, the leaf samples of drought-treated plants were collected separately 15 days after withdrawal of water along with well-watered control and were frozen in liquid nitrogen and stored at −80 °C. Total RNA from the leaves was isolated using Qiagen RNeasy columns (Qiagen, Hilden, Germany). Isolated RNA was then treated with DNase I (Takara) to avoid genomic DNA contamination. The concentration and purity of the isolated RNA were detected by Thermo Scientific NanoDrop 1000 spectrophotometer (Thermo Scientific, Wilmington, Delaware, USA). One μg of DNA-free RNA was used for the synthesis of cDNA using Revert Aid first-strand cDNA synthesis kit (Thermo Fisher Scientific, Waltham, MA, USA).

### 2.4. Quantitative Real-Time PCR (qRT-PCR)

Primers were designed for the selected genes (Appendix A) using online server IDT PrimerQuest (http://www.idtdna.com/scitools/applications/primerquest/default.aspx; (accessed on 20 January 2021)) qRT-PCR assay designing tool. The optimum amplicon size of the assay design was 100, with a GC content of around 50% for the primers, and the probes were designed with an optimum oligo length of 24 bp. The program was performed with 25 μL RT-PCR reaction, including 12.5 μL of the SYBR green RT-PCR master mix (Affymetrix, Santa Clara, CA, USA), 2μL of each forward and reverse primers, 1 μL of cDNA, 1 μL of reverse transcriptase, 0.5 μL of ROX dye and 8 μL of nuclease-free water. Reverse transcription was performed at 50 °C for 30 min and was terminated at 95 °C for 10 min. The PCR reactions included the following thermal cycling conditions: 40 cycles of 94 °C for 3 s, 58 to 60 °C for 1 min, and 72 °C for 30 s.

### 2.5. Identification of Orthologous Sequences and Haplotypes across Millets

Drought genes identified in pearl millet were subjected to find orthologous sequences in other important millet crops having genome sequences such as sorghum (*Sorghum bicolor*), foxtail millet (*Setaria italica*), proso millet (*Panicum miliaceum*) and finger millet (*Eleusine coracana*). Using the local nucleotide BLAST, the sequences were identified with a minimum sequence identity of 85% on average on 50 to 100 N length of the sequence. The e-value was also set to a minimum according to the identified queries. The Orthologous sequences were mapped to the chromosomal location in all millets and represented as a Circos plot using ClicO [29]. The drought gene sequences identified in other millets were fetched out from the genome of the respective crop and used to identify haplotypes of the gene across the millets using [30]. The genes were aligned using ClustalW multiple alignments and visualized for the recognition of the number of haplotypes per gene. 

### 2.6. Computational Protein Analysis

The protein sequences of drought genes were subjected to EMBOSS tool of EBI [31] to find the molecular weight, isoelectric point, charge, and extinction coefficient. Transmembrane helices finding and subcellular location identification were done using TMHMM [32] server (http://www.cbs.dtu.dk/services/TMHMM/; (accessed on 5 February 2021)) and CELLO2GO [33] (http://cello.life.nctu.edu.tw/cello2go/; (accessed on 5 February 2021)). The putative protein sequences were analyzed in MEME, the multiple maximizations for motif elicitation analysis tool (http://meme-suite.org/doc/download.html; (accessed on 5 February 2021)) [34] and SMART (Simple Modular Architecture Research Tool), which allows the identification and annotation of genetically mobile domains (http://smart.embl-heidelberg.de/; (accessed on 5 February 2021)) [35]. The MEME suite was used to search 10 motifs with a base width of a minimum of 6 and a maximum of 60 Kb and an e-value of less than 0.01.

### 2.7. Conserved Sequence Motif across Millets

The collective sequence of orthologous genes from the millets was subjected to finding conserved motifs specific to each gene using the MEME tool [36] with a min 6 N to max 50 N width of each motif. Overall, 14 genes were found to be occurring in all millets. The motifs of these genes identified by MEME software were then directed to TOMTOM [37] to compare them with a known set of motifs in Arabidopsis as a reference. Motif locations and LOGO representation of the motifs were extracted and visualized.

### 2.8. Phylogenetic Analysis

Phylogenetic analysis was done using the protein sequences to understand the evolutionary relationship between the drought genes. To generate the phylogenetic tree, first, the multiple sequence alignment (MSA) was done using CLUSTALW (https://www.genome.jp/tools-bin/clustalw; (accessed on 20 March 2021)) [38] with a gap-open penalty of 10 and gap-extension penalty of 0.05, then the alignment file was imported to simple phylogeny (https://www.ebi.ac.uk/Tools/phylogeny/simple_phylogeny/; (accessed on 20 March 2021)) to form the phylogenetic tree using neighbor-joining (NJ) method. The tree was rendered using iTOL (http://itol.embl.de; (accessed on 20 March 2021)) using a circular rooted arrangement [39].

### 2.9. Gene Ontology (GO)

The genes were submitted to DAVID (https://david.ncifcrf.gov/; (accessed on 20 March 2021)) using *Arabidopsis thaliana* as a reference organism for hierarchical classification into 3 categories: biological process, cellular components, and molecular function [40]. According to that, the genes were classified into different molecular and physiological adaptation groups.

### 2.10. Protein-Protein Interaction (PPI) Network

The STRING version 11.0 (https://string-db.org/; (accessed on 20 March 2021)) was used to create the PPI network of the drought genes and visualize the interaction among the genes [41]. The network was constructed using the confidence score of 0.4 and clusters were created using the k-means value for the categories of different adaptation classes.

## 3. Results

### 3.1. Identification of Drought Gene Orthologs

A total of 171 drought gene sequences from different plant species such as *Arabidopsis thaliana*, *Oryza sativa*, *Zea mays*, *Sorghum bicolor*, *Solanum lycopersicum* were collected from NCBI, TAIR, maizeDB, and RGAP databases. Nucleotide BLAST was performed against pearl millet sequences, resulting in the identification of 92 putative drought genes belonging to the overall group of genes. A large number of gene homologs were found in the molecular (40 genes) and physical adaptation classes (36 genes). A comparatively small number of homologs (5 to 10 genes) were found in ABA signaling, hormone signaling, and detoxification. Due to the small sequence length, some of the sequences were discarded and could not be processed further. Finally, a total of 74 genes were selected for structural and functional experiments.

The 74 drought gene homologs identified in the present study were mapped to all seven chromosomes of pearl millet (Figure 1A). About 60% of genes belonging to different TF families (*bZIP*, *WRKY*, *DREB*, and *MYB*), ion and osmotic homeostasis (*PIP*, *GTG1* and *2*, *KAT*, *PYL9/RCAR1*), and other molecular and physiological classes were mapped on chromosomes 1 and 7. Crucial gene families such as *AnnAt*, *AtBG*, *FSPD*, *RGS*, *DSM*, and *CIPK* were observed in other chromosomes. In the present study, we observed comparatively a smaller number of drought genes were mapped to chromosome 6, whereas chromosome 7 was the hot spot of drought genes. The gene structures analysis revealed that three genes are intron-less while the remaining displayed many introns. The number of exons ranged between 1 (*AP2* and *EDR1*) to 12 (*bZIP*).

The drought genes recognized in the present study were subjected to a BLAST search against other important millet crop genomes, namely sorghum, foxtail millet, proso millet, and finger millet to identify the orthologous genes. The reliability of the orthologous depends on the e-value parameter and hits with a minimum e-value have been considered for further study. The distribution of these genes in pearl millet and other millets are visualized in Figure 1B.

Among all, foxtail millet had the maximum number of matches against the drought genes (85 hits for 42 genes) followed by proso millet (50 hits for 21 genes), finger millet (66 hits for 27 genes), and sorghum (41 hits for 19 genes). The percentage identity varied from 75 to 100 in different crops for identical fragments of the nucleotide sequence. Foxtail millet had 34 genes with 100% identity out of 85 genes with a match length >200. On the other hand, finger millet had 15 genes out of 76 with 95% identity. Sorghum and proso millet had 12 and 13 genes, respectively, with more than 90% identity among the orthologous. 

Some of the genes constitute more than one ortholog in a crop in different chromosomal positions. Among all millets, 19 out of the totally identified 85 orthologous were reported repeatedly in different genome locations of foxtail millet. The gene with a maximum orthologous was TPS1, which was identified in chromosomes 7,8, and 9. Proso millet had the maximum count with 16 repeated orthologous for Myb2, followed by finger millet (13 orthologous for Myb4 and AtBG1), and sorghum (9 orthologous for Myb4 gene).

Drought genes were observed in clusters in different chromosomes of different millets. Among the identified 74 drought genes, 30 were clustered as one group in foxtail millet chromosome 9, followed by 21 in finger millet chromosome 1. Sorghum and proso millet both had 10 to 15 genes clustered in their respective chromosomes 1 and 5.

### 3.2. Haplotypes of Drought Genes

The identified sequences of 74 drought genes in pearl millet were used as a template and fetched out from other millets using nucleotide BLAST. The orthologous drought gene sequences were aligned for the identification of haplotypes across all millets (Figure 2). A total of 698 haplotypes were identified across all genes in different crops. All the haplotypes of pearl millet were also identified in sorghum and proso millet, while foxtail millet and finger millet did not have a few of them. The orthologous sequences of three pearl millet genes (chlorophyll a-b binding protein, bZIP23, and Ann1) identified across the millet crops recorded >90 haplotypes. Nuclear transcription factor Y subunit beta1 (81 haplotypes) was found in all crops except in foxtail millet. It was observed that the MyC2 and Myb4 genes had the minimum number of haplotypes (1 to 3 across crops).

### 3.3. Protein Sequence Analysis

Analyzing the protein sequences provides insights into the characteristic features of the drought-related proteins. The number of amino acids in the proteins ranged from 140 (*NFYB*) to 1510 in (*ABCC*). The isoelectric point value of the protein was also estimated in the range of 3.9 (*CML*) to 12.6 (*DSM2*). The molecular weight of the protein set varies from 15180 to 169079 Dalton. The prediction of the charges of the protein indicated that 40 of them were positively charged, and the remaining were negative. Fifteen transmembrane helices were identified in 20 proteins, while the rest of them had no helices. The location of the proteins in subcellular compartments indicated that 23 of them belong to the nucleus, 21 in the cytoplasm, and 17 in the plasma membrane (Appendix A). Motif identification tool MEME revealed 10 conserved motifs in the protein set, which was distributed in 38 proteins, as shown in (Figure 3). The width of the motifs ranged from 34 to 60 and most of them were present in 3 to 9 different sites of different proteins. Motif numbers 1 to 5 and 8 were conserved, mostly to the kinase families such as *SRK2*, *CDPK*, *CPK*, and *CIPK* with a min e-value of 1.4 × 10^1^, whereas motif 9 was found mostly in the *Gol* family, and motif 6 and 7 in the *PIP* family of proteins. Motif 10 was observed in *bZIP*, *ABF2*, and *CPuORF2* TF genes. The rest of the motifs were distributed in other protein families. Analyzing the protein domains gave rise to 27 major domains belonging to an important family of proteins including *bHLH*, calcium-dependent protein kinase, *MYB* domain protein, and *WRKY* DNA-binding protein (Appendix A).

Gene structures were analyzed to survey the conserved motifs of drought genes identified in pearl millet and their orthologs in the other four millets. In total, 14 genes found in all four crops were subjected to MEME to find the conserved motifs. Among 10 motifs identified, 5 of them (motifs 1–5) were located in C-terminal and 3 (motifs 7, 8 and 10) were located in the middle of the ABO1 gene in millets. The N terminal motifs mainly consist of CC or GA nucleotide for the ABO1 gene. Motifs 1, 3, and 4 were located in the C terminal, motif 2 was consistently present in the middle position, and motifs 8 and 9 were located in the N terminal out of 10 motifs of the *Ann1* gene. *AtBG1* gene had conserved motifs of 1 to 3 in C-terminal for all the millets. *AVP1*, *CML31*, *CDKP7*, *SLAC1*, and *TPS1* genes had a set of respective conserved motifs in their C-terminal. No particular pattern was observed in the rest of the gene sets since the motifs were mostly scattered over the gene in different millets. The LOGO and locational motif representation were visualized in Figure 4.

### 3.4. Phylogenetic Analysis

The phylogenetic tree was constructed using the 74 identified drought genes of pearl millet. The analysis revealed that the genes were separated into five different groups with sub-groups (Figure 5). The genes belonging to ABA signaling (*GTG1* and *2* and *PLY*) were found in the same group whereas BGLU18/AtBG1 was found in a different group. *TFs bZIP23*, *FAR*, *HSTA*, and *ADAP* were clustered in Group 4, and *MYB*, *NF-Y*, *WRKY*, *NCED*, *DREB*, and *SNAC* were in Group 3. Ion and osmolyte factors such as *ATHB*, *GPA1*, *DSM*, *OST1*, and *OST2* were in the same cluster (Group 3) except *CPK*, *RBOHF*, and *PIP* family of genes. Hormone and signal transduction genes *NCED*, *CYP*, *AAO*, *CIPK*, and *CML* were scattered in different groups. Genes related to detoxification (*GolS1*, *GolS2*, and *P5CSA*) were found in the same subgroup of Group 5. Other important genes in the class of molecular and physiological adaptation such as NF-YB, RGS1, PIP, and CDPK were grouped into another sub-group of Group 5.

### 3.5. GO Analysis

By analyzing the 74 genes in DAVID for GO annotation, 28, 16, and 3 GO terms were identified in different biological processes, molecular functions, and cellular components, respectively. The annotations involved in biological processes were water deprivation, ABA-activated signaling pathway, salt stress, jasmonic acid, heat, ABA catabolic process, protein de-phosphorylation, cold, salicylic acid, desiccation, stomatal movement, sugar mediated signaling pathway, galactose metabolic process, positive regulation of transcription, de-phosphorylation, sterol metabolic process, regulation of stomatal movement, carbohydrate biosynthetic process, high light intensity, and oxidative stress. Almost 30% of genes are involved in molecular functions, including transferase activity, transferring glycosyl groups, transcription regulatory region DNA binding, sequence-specific DNA binding, DNA binding, TF activity, sequence-specific DNA binding, and protein binding. Cellular components had GO annotations such as protein serine/threonine phosphatase complex and cytoplasmic components (Appendix A).

### 3.6. Differential Expression of Drought Genes

A set of 48 pearl millet inbreds were exposed to drought stress and the differential expression of 74 drought-responsive genes across four maturity groups were studied at the flowering stage. These genes were categorized into eight functional groups and the expression pattern in the control and drought-induced genotypes is furnished below.

### 3.7. ABA Signalling

All four genes (*AtBG1*, *GTG1*, *GTG2*, *PYL9/RCAR1*) responsible for ABA signaling showed a high level of up-regulation in ICMR 100948 belonging to the very-early maturity group (Figure 6A). Similarly, positive regulation was exhibited by all the four genes in ICMB 03999 with the magnitude of expression being higher in *PYL9/RCAR1* when compared to the other three genes. *PYL9/RCAR1* in ICMB 100391 was highly down-regulated by a 2.3-fold change over control, whereas other genes showed positive expression. For ICMB 00111, all genes were positively expressed, except *GTG2*, which possesses a 2-fold decrease in expression than control under drought stress. While all four genes showed variable expression in genotype ICMB 15222, with *GTG2* being highly up-regulated, *PYL9/RCAR1* and *GTG1* with positive expression, and *AtBG1* showing a two-fold decrease in expression rather than control.

Early flowering genotypes ICMR 100229 and ICMB 1608 had an equal level of over-expression in all four genes, with *PYL9/RCAR1* showing more up-regulation in ICMB 1608 (Figure 6B). In ICMP 100443, up-regulation was observed in all the genes except *AtBG1*, which was negatively regulated under stress conditions. Down-regulation of four genes was noticed in ICMB 92888, and *GTG2* showed maximum down-regulation in most of the genotypes (ICMB 100270, ICMB 11999, ICMR 101221, ICMB 100673, and ICMB 92888). *GTG1* and *GTG2* were highly down-regulated in ICMB 11999, while the other two genes showed moderate positive regulation. The medium-maturing genotype, ICMR 100999 had up-regulation for all the genes with *ATBG1* being over-expressed nine times more than the control, whereas in ICMB 100637, it was least positively expressed when compared to the other three genes (Figure 7A). *GTG2* expressed a higher level of down-regulation in ICMB 100642 and ICMR 101011 with a four- to five-times decrease in expression than control. Similarly, *PYL9/RCAR1* was highly down-regulated in ICMR 101011 and ICMR 100028 under drought conditions.

*AtBG1* and *PYL9/RCAR1* showed high up-regulation in most of the late-maturing genotypes when compared with the other two genes (Figure 7B). Up-regulation of all four genes was observed only in ICMR 100152 and ICMR 100544. In ICMR 100045, *GTG1* and *PYL9/RCAR1* were up-regulated, whereas down-regulation was observed in *AtBG1* and *GTG2*, in which *AtBG1* showed a negative expression that is around five-fold less than control. A high level of up-regulation was exhibited by *AtBG1* and *GTG2* in ICMB 100252, where *GTG1* showed a five-times decreased expression rather than control. Similarly, in ICMB 100619 both *GTG1* and *GTG2* showed a three-times decrease in the expression, whereas *AtBG1* and *PYL9/RCAR1* were positively expressed.

### 3.8. Hormone Signalling

Under drought conditions, in the early maturity group, a high level of differential expression of genes, controlling hormone signaling was observed compared to the mid and late maturity genotypes. The extent of up-regulation of all four genes (*AAO3, CYP707A1, CYP707A3, NCED*) was maximum in ICMB 00111, with a high level of positive expression exhibited by *CYP707A3* with a five-fold increase in expression rather than control. Except for *AAO3* and *NCED*, the expression of *CYP707A1* and *CYP707A3* was very high in ICMB 15222 (very-early maturity group) (Figure 4). When compared to other genes, *NCED* expression was three-fold lesser in ICMB 03999, but an opposite expression of this gene was found in ICMR 100948 with 40-times more up-regulation than control. In the same genotype, *AAO3* and *CYP707A1* showed a two- to three-times decrease in expression. ICMB 1608 and ICMB 100649 belonging to the early-maturing group exhibited an equal level of up-regulation of all genes (Figure 5). Most of the genes got down-regulated in ICMB 100270, ICMB 100673, and ICMB 92888. *CYP707A3* was over-expressed in ICMP 100443, whereas the other genes such as *CYP707A1* and *NCED* showed five-times lesser expression under stress rather than under control.

The medium-maturing group had a moderate level of expression of the studied genes, with *AAO3*, *CYP707A1*, and *CYP707A3* showing positive expression and *NCED* with negative regulation in ICMR 100999 (Figure 7A). Expression of *AAO3* was more down-regulated in many genotypes when compared to other genes, with a 2.5-fold decreased expression rather than control in ICMB 100637, and ICMR 101011. In ICMB 100638, which belonged to the late-maturity group, all genes were expressed at a higher level except for *NCED*, which showed negative regulation that was approximately five times lower than control (Figure 7B). The magnitude of expression of the *CYP707A3* gene was 36 times higher than control in ICMR 100045, whereas other genes were three to five times lower in their expression under drought stress.

### 3.9. Ion and Osmotic Homeostasis

Sixteen genes were differentially expressed targeting ion and osmotic homeostasis, with a maximum level of up-regulation of genes observed in ICMB 03999, which belongs to the very-early maturity group (Figure 6A). ICMB 00111 also showed a comparatively higher level of expression in all genes except in *OST2, PIP2;5*, and *SLAC1. OST1/SRK2E* was highly up-regulated in the genotypes of the early-maturing group, with a maximum up-regulation observed in ICMB 15222.

ICMR 100229 was another genotype in the early-maturity group to show up-regulation for most of the genes with a maximum expression by *PIP2;5* (Figure 6B). Most of the genes down-regulated in ICMB 11999, of which *SLAC1* showed a three-fold decline over control. ICMB 100649 also showed a high magnitude of down-regulation for *PEPCK* and *DSM2*.

Among all 16 genes, more down-regulation was shown by *BnPIP1* in ICMR 101011, followed by ICMR 100028 in the medium-maturity group (Figure 7A). A maximum level of up-regulation was shown by *PIP1;4* in ICMB 100637 with a 12 times higher expression than in control. Genotypes in the late-maturity group had the maximum up-regulation of *OST2* with a 22 times higher expression than control in ICMB 100619 (Figure 7B). The same genotype showed down-regulation in seven genes, and the least expression was explained by *ABCG40* which was 6-times less than the control. A high level of positive expression of *PIP1;4* (32-times) more than control was observed in the late-maturing genotype, ICMR100544, when compared to other genes.

### 3.10. TFs-Mediated Regulation

Twelve genes were identified to be involved in TFs-mediated drought control and the expression of all genes was very high in ICMB 00111, where *HSFA1b* showed eight times increased expression rather than control. In the very early maturing genotype ICMB 03999, *ADAP* had the highest expression value of 88-fold under drought (Figure 6A). ICMR 100948 also had a significant level of positive expression of genes, except for *AREB1* and *ADAP*. In the early-maturity group, among the 12 genes, 10 were negatively expressed in ICMB 100270, while only *ADAP* and *AtNF-YB1* showed positive expression (Figure 6B). *AREB1* showed a maximum expression 87-fold under drought in early-maturing genotype ICMR 100229, while *HSFA1b* and *ADAP* were the two genes with down-regulation.

*MYB60* was the one expressed higher-level when compared to the rest of the genes in medium-maturing genotype ICMB 100637, with 82-fold under drought conditions (Figure 7A). Positive regulation of all the genes was observed in ICMR 100999, whereas down-regulation of those genes was recorded in ICMR 101011 and ICMR 100028, with the highest negative expression showed by *HSFA1b* and *AtNF-YB1*. *MYB60* and *FAR1* were the important TFs, highly up-regulated in the late-maturity group genotypes, ICMB 100635 and ICMR 100045. *AtNF-YB1* was the only gene with five times more down-regulation over control in ICMR 100544 (Figure 7B).

### 3.11. Molecular Adaptation

The expression of all 16 genes involved in molecular response favoring drought tolerance was positively regulated in ICMR 100948 and ICMB 15222, with a high level of expression shown by *SHINE1* (*SHN1/WIN1*) followed by *OsWRKY45*. In the early maturity group, maximum positive expression was explained by *OsWRKY45*, with 87-fold in ICMR 100229 followed by *FSPD1* in ICMP 100443 having 84-fold (Figure 6A).

Except for *CIPK15*, all other 15 genes showed positive regulation under stress conditions in medium-maturing genotype ICMB 100637 (Figure 7A). On the other hand, in the same group, ICMR 101011 showed complete down-regulation for all genes except for *OsiSAP8*, *OsDREB1A*, and *AP37*. ICMR 100544, belonging to the late-maturity group, showed a clear-cut up-regulation in all 16 genes, with a maximum expression by *AP37* and *FSPD1* more than 80-fold change when compared to control (Figure 7B). An increase in the level of expression of the *PFA-DSP1* gene was noticed in the late-maturity group genotypes and it was more than 89-fold in ICMB 100252.

### 3.12. Signal Transduction

All 11 genes associated with the signal transduction process had positively regulated in ICMR 100948. *CML9* was an important gene, showing 45-times increased expression in ICMB 03999 over control. ICMB 15222 also had positive regulation of most of the genes, except for *ABI1* and *GbRLK*.

Even though most of the genes had high negative regulation, *ABI2* and *CIPK12* were highly up-regulated in the early-maturity group genotypes (Figure 6B). ICMR 100999 belonging to the medium-maturity group had the highest expression for *RGS1* followed by *CPK23* with 86-fold and 81-fold, respectively, over control. *RGS1* showed positive regulation in ICMB 100637, a genotype that belongs to the medium maturity group (Figure 7A). A high degree of up-regulation was showed by *SRK2C* and *CPK23* in ICMB 100252 and ICMB 100619 (late-maturity group), while other genes showed moderate expression (Figure 7B). Although the expression was not too high, ICMR 100152 showed positive expression for all 11 genes involved in stress response.

### 3.13. Physiological Adaptation

All five genes (*AtrbohD*, *EDT1/HDG11*, *GPA1*, *HAB1*, and *KAT2*) selected for studying physiological adaptation showed up-regulation in ICMB 03999 belonging to the very early maturity group, of which *AtrbohD* showed the highest level of up-regulation, with 16 times more under drought than under control (Figure 6A). ICMB 15222 too had a high level of differential expression of genes, except for *EDT1/HDG11*, which was down-regulated three-fold when compared to control. *AtrbohD* and *HAB1* showed negative regulation in the very early maturing genotypes, ICMB 100663 and ICMR 100948, with a 10- to 11-times decrease over control.

*KAT2* was highly expressed (44 times) over control in ICMP 100443 (early maturity group) (Figure 6B). In ICMP 100443 and ICMR 100229, all genes were up-regulated under drought conditions. In ICMB 100673, except for the *GPA1* gene, all other four genes showed negative expression when compared with the control. The expression level of *EDT1/HDG11* reached 14 times more than the control in the late-maturity genotype ICMB 100638, along with the up-regulation of *HAB1* (Figure 7B). Except for *HAB1*, the other four genes showed negative expression in ICMB 100619.

### 3.14. Detoxification

All six genes (*ERD1*, *GolS1*, *GolS2*, *MYB4*, *P5CS1*, and *TPS1*) were positively regulated in ICMR 100948 (very-early maturity group), with a significant level of expression of *TPS1* (87-fold), followed by *ERD1* (80-fold) (Figure 6A). *ERD1* expression was high (80-fold) in very-early maturing genotypes ICMR-100948 and ICMB-03999. Maximum down-regulation of genes was observed in ICMB 00111 and ICMB 100663 except for MYB4 and TPS1 under stress conditions.

When compared to other genotypes, significant up-regulation in all drought-responsive genes was observed in ICMR 100152. The level of expression of *ERD1* and *GolS2* genes was more than 40 times in stress when compared to the control condition in the late-maturing genotypes, ICMB 100252 and ICMR 100544. *P5CS1* also showed comparatively a high level of up-regulation in the late-maturity groups (Figure 7B). Most of the drought genes were up-regulated in ICMB 100252, except the *GolS2*, which was seven times lesser than the control.

### 3.15. PPI Network

The analysis of PPI revealed a cross-linked network among the 74 identified drought-responsive genes in pearl millet. The functional groups were labeled using different colors (Figure 8). The degree of connectivity of nodes varied from 1 to 44, with an average clustering coefficient of 0.541. Two mostly connected genes with a value of 44 degrees were *ABI2* and *P5CS1*. Other hub genes *ABI1*, *OST*, *NCED*, *CDPK*, *DREBA*, *HAB*, *MYC*, *PIP*, and *GTG* had connectivity between 20 to 40 and belong to different functional classes such as TFs, signal transduction, and ion and osmotic homeostasis. Among all TFs, *MYC* and *MYB* were mostly inter-connected with other genes belonging to various molecular and physiological functions.

## 4. Discussion

Drought is one of the major constraints which affects grain and fodder productivity worldwide. Pearl millet is widely grown in arid and semi-arid regions which makes it prone to stress induced by drought. When compared to other cereals, pearl millet shows significant tolerance to drought, due to certain molecular and genetic mechanisms operating within the crop. Therefore, understanding the mechanism of drought tolerance and the development of drought-tolerant varieties are the key strategies to improving the yield under drought-prone ecologies.

### 4.1. Gene Orthologs

In the present study, we collected 171 drought-responsive genes from different crops, of which 74 genes were selected for structural and functional studies. Some homologs were not considered due to the short sequence length and dissimilarity in regions when compared with the original gene nucleotide sequence. The variation observed in the sequence composition and the length of the nucleotide between the drought genes of pearl millet and the original gene sequence indicated the differential divergence during the process of evolution [42]. The distribution of drought-responsive genes in all seven chromosomes of pearl millet varies in respect to the presence of the gene or the clustering of the genes in the chromosomal region. Among other chromosomes, Chromosome 7 was the hot spot, with a cluster of the important gene families. Drought QTLs distributed in different chromosomes of pearl millet were identified in the multi-location experiments [43,44].

Orthology analysis generalizes the assumption that genetic material is propagated by vertical descent from pre-existing genes by speciation [45] Orthologous sequences of the 74 important drought genes of pearl millet were identified through an in-silico approach. The millet crops with existing reference genomes such as sorghum, foxtail, proso, and finger millet were taken for the identification of orthologous genes.

The orthologous sequences of the 74 important drought genes of pearl millet were observed to be distributed in various chromosomal positions in different crops. By comparing the genomic properties of different millets, a varying level of evolutionary distances was found among them [46]. The maximum number of orthologous sequences were identified in foxtail millet with the maximum identity, whereas the least number of gene orthologous were identified in sorghum among all millets. More number of orthologs indicated the close evolutionary relationship between the set of studied genes. Evolutionary divergence among gene families is greater because of the environmental factors, selection, and genomic rearrangements. Identification of orthologs among millets can provide functional insights into the genes and phylogenetic inference. The experiment also identified the common genes and SNPs which could be used across millet crops to develop strategies to combat drought as most of the millets grew in similar marginal ecologies.

### 4.2. Drought Gene Haplotype Identification in Millet Crops

The haplotype is a collection of specific alleles in a cluster of tightly linked genes that are likely to be inherited together; hence, they are likely to be conserved as a sequence [47]. The genes bZIP, Ann1, chlorophyll a-b binding protein, and AtBG1 identified across the millet species showed highly conserved regions with a high number of haplotypes. These are the significant gene strongly conserved at the nucleotide sequence level throughout the related genomes. The significant number of haplotypes observed in 23 pearl millet genes with other millet crops probably indicated the conservation of these sequences during the evolution process. Myb 4 and Myc showed the highest level of variability or polymorphism by recording few numbers of haplotypes which indicated the several events of mutation and recombination in the gene sequences during the process of evolution. The continuous series of haplotypes detected for the drought genes across the millets could be used for designing the common markers for its utilization across the crop species.

### 4.3. Conserved Motifs and Domains

Motifs play an important role in understanding transcriptional regulation. The similarity and diversity of protein motifs belonging to different gene families provide insights into the structure and function of the protein. To understand the functional characteristics of the gene by specific and conserved peptide sequences, 10 motifs were identified in the 74 drought-responsive genes. Motifs one to five and eight were mostly observed in the calcium-dependent kinase family (*CDPK*) of protein. Plant *CDPK* plays an important role in abiotic stress tolerance [48,49]. They are affected by the calcium level of the plant which works as a secondary messenger. The concentration of calcium can be rapidly disturbed by an increase in hormones, light, and abiotic stress [49,50]. The *CDPK* family of genes is predominant in intracellular Ca^2+^ influx and phosphorylation [50,51]. The protein kinase family of domain consists of the ATP binding domain and is adjacent to the auto-inhibitory domain. The binding of calcium to the auto-inhibitory domain changes the protein conformation [52]. The Interaction of the auto-inhibitory domain with the calcium-binding domain regulated the *CDPKs*, *CaMKs*, and *SnRKs* [53,54,55]. Calcium-dependent kinases were found in various plants; for example, 34 *CDPK* genes were found in Arabidopsis, 31 in rice, 40 in maize, and 20 in wheat [56,57].

The second most evident motifs six and seven were mostly observed in the *PIP* family of proteins. The plasma membrane intrinsic proteins correspond to aquaporins that are expressed in plasma and vacuolar membrane and intracellular water transport [58]. Aquaporins have a secondary structure commonly observed with six transmembrane alpha-helicases connected with five loops, of which two are hydrophobic [59]. The *PIP* family is the largest subfamily of plant aquaporin, and it is divided into *PIP1* and *PIP2* groups. The amino-terminal extension is shorter and the carboxy-terminal is longer in *PIP2* than *PIP1* isoform [60,61]. The activity of *PIP2* can be enhanced by *PIP1* proteins [62]. Motif nine was observed in the *Gol* family of genes *GolS1* and *GolS2*. Galactinol Synthase is one of the key enzymes in metabolic pathways leading to the biosynthesis of RFOs. *GolS* perform galactosylates reaction to convert myo-inositol to form O-α-d-galactopyranosyl- [1 → 1]-l-myo-inositol, which is commonly known as galactinol. The sequential transfer of α-galactose from galactinol onto sucrose yields raffinose and stachyose [63]. Research shows that raffinose content increases upon heat stress. The expression of *GolS1* and *GolS2* in Arabidopsis is regulated by a *heat shock transcription factor* (*HSF*) [64]. A set of candidate genes associated with heat-responsive pathways activating heat shock factors have been identified in common beans [25]. Motif 10 was observed in *bZIP*, *AREB*, and *CPuORF2* proteins. *bZIP* is a basic leucine zipper transcription factor that plays various roles in plant growth and stress responses [65]. Approximately 100 *bZIP* families identified in *Arabidopsis thaliana* were classified into sub-families depending on their structure and function [66]. *ABF/AREB*, from a subfamily, was reported in participating in ABA signaling in response to abiotic stresses [67,68].

Being staple and climate-resilient crops, comparing sorghum, foxtail millet, proso millet, and finger millet drought gene sequences with pearl millet provides the knowledge on the gene relatedness and evolutionary history shared among them. From the drought genes comparison, it appears that only foxtail millet is very close to the pearl millet. Though the millets might have diverged in varying degrees from each other, similarities in their drought gene structure were conserved to a significant extent, which was proved by the motif similarities. The conserved positioning of similar motifs in the N or C terminal of a gene among different millets states that gene families were synchronous.

The domain is an evolutionarily conserved unit in protein, a combination of motifs gives rise to a particular functional domain in the protein. Twenty-seven conserved domains were observed in the protein sequence set related to drought. The abundant ones were ABA-related protein domains. ATPase associated with various cellular activities (*AAA*) protein is an important regulator in diverse cellular activities. One of the protein homologs *SKD1*, which encodes *AAA* in *Zea mays*, was identified to be involved in salt or drought stress tolerance [69]. *Annexins*, an ANX domain-containing protein, is reported to play an important role in plant stress responses. In Arabidopsis, *AnnAt1* was up-regulated in stress-induced leaves and over-expression of *AnnAt1* imparted drought tolerance in the plant [70]. Plant Ca^2+^ATPases is an important domain, expressed during heat, drought, and abiotic stresses, and the interaction of genes with other developmental factors was studied in *Triticum aestivum*. Ca^2+^ATPases genes consist of two distinct Auto-inhibited Calcium ATpase (ACAs) and Endoplasmic reticulum Calcium ATpase (ECAs) groups that pump calcium outside the cytoplasm during homeostasis [71]. *Caseinolytic Protease B* (*CLPB*) proteins, a high molecular weight chaperon, are a part of the HSP 100 family. The protein has a two-tiered hexamer ring connected with coiled–coiled linkers. *CLPB* was also classified as a member of the *AAA* (*ATPases Associated with diverse cellular Activities*), ATPase superfamily. *CLBP* in *Arabidopsis thaliana* plays an important role in plant survival, while ortholog in tomato was known to be important for heat acclimation [72].

*Apetala2/Ethylene responsive factor* (*AP2/ERF*) family of TFs exhibit as a key regulator in plant response to stress by activating ABA and Ethylene (ET) and independent stress-responsive genes [73]. In Arabidopsis, different *MYB* TFs were characterized for their drought responses [74]. The *MYB* TF is characterized by the presence of the MYB domain, classified based on repeated sequences containing 52 amino acid residues, and forms an alpha-helix, which is involved in DNA binding [75]. *DREB* (*dehydration-responsive element binding*) functions by binding with the DRE/CRT *cis*-elements in the promoter region [76]. The function of *DREB* was reported to be regulated highly in drought conditions through the expression of stress-induced genes, mostly in an ABA-independent manner [26,77]. Among all the TFs involved in the life process of plants, *WRKY* is a very important one, playing a role in physiological processes such as growth, metabolism, and response to various stresses. Over-expression of *WRKY* increased the drought tolerance in *Medicago truncatula* [78].

The *bHLH* proteins are the conserved family of TF that are involved in plants to cope with drought stress through stomatal development, root hair formation, and hormone metabolism [79]. The *G-protein-coupled receptor* proteins (*GPCR*) are observed as the largest and most diverse family of membrane proteins that enhance drought tolerance in transgenic plants by promoting root growth and induction of ROS-scavenging enzymes [80]. *GPCR* proteins are mainly structurally similar but differ in their receptor sites, and one common feature is the presence of 7-TM helices [81]. In *Arabidopsis thaliana*, two members of the *GPCR* family were found, such as *GTG_1* and *GTG_2*, to be novel types of GPC-GTPases, which function as abscisic acid (ABA) receptors [82] and regulate drought tolerance.

*KAT* protein is involved in the potassium channel, which is inevitable for plant development and growth. Seventy-one K^+^ transporter proteins were identified in *Arabidopsis thaliana* and two inward rectifying channels *AKT1* and *KAT1* were identified as related to their behavior under stress [83,84].

### 4.4. Phylogenetic Relationship among Drought-Related Genes

Phylogenetic analysis revealed the evolutionary history of genes and species. The genes belonging to *MYB*, *WRKY*, *DREB*, *NAC*, *FAR*, *HSFA1b*, *bZIP*, and *ADAP* were mostly bifurcated into two groups. Usually, similar functions were performed by the genes which have close evolutionary relations [42]. (A close relationship among the TFs and ion, osmolytes genes indicated their synergistic operation in plants under drought conditions. Same gene families have multiple functions which regulate different mechanisms to improve drought tolerance in pearl millet as well as in other species. Genes related to hormone signaling such as *NCED*, *AAO*, and *CYP70* were related to signal transduction genes of *SRK2*, *CML*, *AB2*, and *CDPK*. The clustering of one group of the gene (such as TFs) with different functional genes indicated the interaction of the genes in molecular or biological processes. Mixed clustering of genes based on their sequence homology and paralogy correlates with the function of the gene [85]. During the evolutionary period, among the multigene families, the structure of the genes was commonly diversified, which facilitates the adoption of new functions according to the changing environment [86]. Closely related ortholog genes *bZIP* and *PIP* were observed in the same subgroup, suggesting that ancestral genes existed before the divergence.

### 4.5. Expression Analysis of Genes under Drought Condition

In our study, by exposing 48 pearl millet inbreds to drought stress, the differential expression of 74 genes distributed in eight functional groups was studied. Among the genotypes studied, ICMR 100152, ICMR 100544, ICMB 03999, and ICMB 100229 showed the maximum number of up-regulation in 69, 63, 61, and 61 genes, respectively. ICMR 101011, ICMB 100673, and ICMB 100270 showed the highest number of down-regulation in 51, 48, and 45 genes, respectively. Genes *PYL9/RCAR1*, *CPK23*, and *OST1/SRK2E* had maximum positive expression in 41, 40, and 38 genotypes, respectively, while genes *MRP4*, *AAO3*, *AVP1*, and *OSMSR2* genes were highly negatively expressed in more than 21 genotypes (Table 1).

Depending on the participation in various pathways genes are mainly divided into two categories ABA-dependent and ABA-independent [8]. Regulatory proteins, such as TFs, protein kinases, and ABA biosynthesis, are in one group and functional proteins such as water and ion channel, detoxification, enzymes, and proteins involved in osmolyte biosynthesis are in the second group [87].

Plants deal with drought by activating complex signaling networks that produce physiological responses. Signal transduction in ABA-dependent pathways increases with ABA concentration, which is sensed by receptors like *GTG1, GTG2, and PYR/PYL.* Once ABA binds to the *ABI* protein, the phosphate activity gets blocked by auto-phosphorylation and activation of *OST1*, a serine-threonine kinase in open stomata. *OST1* and *CDPK* activate the *ABA-responsive element-binding* proteins (*ABFs/AREBPs*) and *SLAC*, respectively, under stress conditions [88]. GO annotation indicated that almost 57% of genes were involved in drought-related responses, of which ABA catabolic process, protein serine/threonine phosphatase complex, inositol 3-alpha-galactosyltransferase activity, and GTPase-binding activity were highly enriched. In our study, *GTG2* was up-regulated five-fold in ICMP 100443. *GTG1/GTG2* is the ABA-binding receptor protein that plays a crucial role in the signaling mechanism [89]. *BGLU* in rice is localized in chloroplast and affects the cellular ABA pools in response to drought stress [90]. *AtBG1* was up-regulated in ICMB 100252 and slightly down-regulated in the ICMR 100045. We found *CYP707A* gene was highly up-regulated in ICMB 00111, ICMB-15222, ICMB-100443, and ICMR-100045, which emphasized the CYP707A involvement in ABA signaling, dehydration, and rehydration condition [91].

*NCED* was up-regulated in the ABA biosynthesis pathway, which acts as a rate-limiting factor for ABA [92,93]. Drought stress treatment in maize, tomato, bean, Arabidopsis, cowpea, and avocado showed high expression of *NCED* [94]. ICMR 100948 belonging to the very-early maturity group showed a higher expression level for the *NCED* in pearl millet.

Early-maturing genotype ICMR 100229 had a higher expression of *WRKY45*. On exposure to drought and ABA treatment, up-regulation of *TaWRKY1* was observed in tobacco, resulting in drought tolerance via stomatal closure and altered osmotic adjustment to accumulate higher biomass [95]. TFs *FAR1, SNAC1, MYB60, AREB1, and ADAP* were highly up-regulated in ICMR 100045, ICMR 100544, ICMB 100638, ICMR 100229, and ICMB 03999, respectively. *AtMYB60* was characterized for its role in stomatal movements, as well as in lateral root growth in Arabidopsis. *AtMYB60* expression in roots was induced by auxin, and the over-expression resulted in increased water uptake during drought stress [96]. In rice, over-expression of *SNAC3*, a stress-related TF, increased the heat tolerance by decreasing the *H2O2* and *MDA* content in leaves, lowering ion leakage, and inducing many ROS-associated genes which enhanced the cell membrane stability and redox homeostasis. It also enhanced the drought tolerance by reducing the water loss and is involved in osmotic adjustment [97]. A constant positive regulation was observed for *SNAC* TF in many genotypes under drought stress, of which ICMR 100544 had a higher expression. Under drought stress, a high-level accumulation of *SNAC1* was observed in guard cells in rice. Its over-expression resulted in reduced transpiration loss due to stomatal closure [98].

*FAR1* is an important TF up-regulated in ICMR 100045, which led to the deactivation of cell death and decreased the accumulation of ROS under drought stress [99]. *ADAP* was up-regulated in ICMB 03999 and has a similar function to *ARIA*, which is a positive regulator of ABA response. *ADAP* interacts with other TFs such as *bZIP* and *ABF2*, regulating the seedling growth [100]. HSPs have different regulatory patterns at the transcriptional level. HSPs act as protectors of cellular proteins from damage under drought conditions [101]. A study conducted in common bean has identified a set of 22 heat-responsive candidate genes involved in the activation of heat-shock proteins (*MED23*, *MED25*, *HSFB1*, *HSP40*, and *HSP20*) related to thermostability in plants and heat-responsive signaling pathways via abscisic acid and auxin [25]. *HSFA1b* was highly up-regulated in the very-early maturing genotype ICMB 00111.

ICMP 100443 showed up-regulation of *FSPD1* in pearl millet, which manages the levels of spermine in leaves and roots. Higher expression of *FSPD1* was found to enhance the drought tolerance in sweet potatoes [102]. *A1SAP* expression level in ICMB 100638 was high, which regulates the leaf-rolling under drought conditions and results in a higher accumulation of green biomass during vegetative growth, along with maintenance of productive tillering and grain filling [103]. ICMR 100999 showed higher regulation of *RGS1*, which changes the expression of many genes involved in ABA biosynthesis, leading to changing the stress responsiveness [104].

*AtTPS1*, a key gene in trehalose biosynthesis, was up-regulated in the very early maturing genotype ICMR 100948. TPS1 in transgenic maize showed better responses to drought stress through an increased accumulation of trehalose, decreased stomatal density, and a reduced rate of transpiration [105]. ICMR 100229 had high up-regulation for *ABA-responsive element-binding protein (AREB)/ABFs* (*ABRE binding factor*) which was found to modulate the gene expression during ABA signaling under osmotic stress conditions [106]. The over-expression of *AREB1* from Arabidopsis showed drought tolerance in soybean [107]. Over-expression of wheat TF *TaAREB3* has enhanced ABA sensitivity and drought tolerance in Arabidopsis [108]. The TF *FAR* involved in carotenoid biosynthesis in shoots conferred drought tolerance by up-regulating in chickpea [109].

A complete microarray expression study in Arabidopsis revealed that *Early responsive to dehydration stress 1* (*ERD 1*), which encodes Clp protease and functions with *ZFHD1*, and *NAC* improves the drought tolerance [110]. Evidence from our study explained that *ERD1* was positively regulated in ICMB 100252.

The functional proteins such as *OST2*, *ABO1/ELO1*, *PIP1*, *ATHB6*, *SRK2*, *PEPCK*, and *DSM2* in the ion and osmolytes category were highly up-regulated in ICMB 10061, ICMB 100252, ICMR 100544, ICMB 03999, ICMB 15222, ICMB 100649, and ICMB 100650, respectively. ICMR 100544 has shown higher expression for *PIP*, and the activity was mostly responsive to kinases and transferases involving signal transduction, cellular processes, and localization of molecules. In *Macrotyloma uniflorum*, the expression of *PIPs* in shoot and root led to drought tolerance [111].

A high level of positive regulation of *GolS2* was observed in ICMR 100544. The expression of genes involved in osmoprotectant biosynthesis changes in drought stress conditions, particularly in the roots. Raffinose and galactinol are involved in tolerance to drought and high salinity stress. A study in Arabidopsis indicated that *GolS1* was up-regulated in shoot and *GolS2* was up-regulated in root under drought [10]. A significant up-regulation of *ATBH* in ICMB 03999 has a role in the regulation of the cell division in developing organs under stress [112]. *SnRK2* up-regulated in ICMB 100252 was also activated by osmotic stress in Arabidopsis in concert with up-regulation of *DREB1A* under drought stress [113]. The difference in the pattern of nucleotide variation among the *DREB2* genes in wild and cultivated common beans has contributed to the variation in the level of tolerance to drought [26]. Ketoacyl-CoA thiolase (KAT), an important enzyme in fatty acid degradation was highly regulated in ICMB 100443. The over-expression of *KAT2* in transgenic lines showed ABA-induced stomatal closure and inhibition of stomatal opening under drought conditions [114].

### 4.6. Interaction of Drought-Related Genes in PPI Network

In plants, proteins work in a network to control the trait expression [115]. PPI is an important component to understand the gene regulation and mediation of most cellular processes [116]. The interactions among different genes are the salient source for estimating the diversity in the gene expression pattern [117]. Genes involved in many biological processes, such as cell cycle growth, development, and response to environmental stresses, are inter-connected based on their regulation. Major TFs, control the gene expression based on condition-specific responses [117]. The PPI network was constructed using 74 drought-responsive genes belonging to different functional and regulatory classes. The network (Figure 8) demonstrated that a large proportion of genes identified in molecular adaptation and TFs were connected with other categories of genes, while genes residing in ABA signaling, hormone signaling, ion and osmolytes, and signal transduction categories were clustered together. *ABI2* and *P5CS1* were the two hub genes identified with the degree of 44 connectivity, interacting with all the major molecular adaptation factors, and ABA signaling proteins. According to the network, many proteins were identified as semi-hubs, with a connectivity of 20 to 30.

The clustering coefficient gave rise to the group of genes cross-talking with each other in response to related functions. ABA is a key regulator of seed maturation, germination, and adaptive responses to stress conditions in the environment. *ABI1* and *2* are ABA-insensitive genes that encode type 2c protein phosphate with redundant yet significant functions. In Arabidopsis, *ABI* encodes a serine/threonine phosphate protein, which contributes to ABA signaling and regulates the ABA responsiveness [118]. The connectivity of different TFs such as *DREB*, *AGB1*, and *MYC* with *ABI1* and *ABI2* can infer the regulation of this gene under stress-induced conditions. Proline gets accumulated and functions as an osmo-regulator under drought stress as explained in sugarcane, where a significant amount of proline production was observed under drought [119].

Our network analysis highlighted the positive interaction of osmotic stress, drought, and cold stress on *P5CS1* and the regulatory role of *MYB2*, *ERF-1*, and *EIN3* TFs in wheat by an accumulation of proline [120]. It also identified *ABRE* and *DREB*, as two important nodes with 36 connections in the network, and both are cis-regulatory elements that function in ABA-dependent and ABA-independent manner with high expression in response to freezing, drought, and salt stresses in *Arabidopsis thaliana* [121,122]. TFs belong to *ERF/AP1* family that bind to *DRE/CRT* elements and are termed as *DREB* [123]. In common bean, among the two *DREB2* gene families, *DREB2A* has more significant molecular variations than *DREB2B* with respect to geographical origin [26]. Mostly *ERD, NAC, MYC, and NCED* TFs, and other genes such as *OST, GolS, HAI, and PIP* were interacting with *DREB. MYC* and *MYB* are the two TFs to bind cis-elements in the promoter sequence. Over-expression of *MYC/MYB* in transgenic plants resulted in improving the osmotic stress tolerance of the plants [124] Many *MYB* TFs were the hub-gene with degree 26 and were involved in a range of molecular and physiological processes including developmental control, cell fate determination, stress responses, hormones, signal transduction, and pathogen defense. The interaction of the *MYB* with *WRKY*, *PYLR*, *PSAG*, *OST*, and *NCED* indicated the inter-relatedness among the TF. Around 65% of *MYB* genes expressed in rice (*Oryza sativa*) seedlings expressed differentially under drought stress [125]. Experiments also found that 51% and 41% of *MYB* genes were up and down-regulated, respectively in Arabidopsis under drought conditions [126].

*OST1/SRK2E* is an ABA-activated protein kinase that functions by controlling the signal transduction pathway of stomatal closure [127,128]. *OST* was associated with *RCAR, OZS, RBOHF, SNRK, PYL, PCS, PIPA, PYR, and RBOHD* protein families in the network. Arabidopsis and rice contain 10 members of the *SNF1*-related PKase family. The activation of *OST1* also depends on osmotic and drought stress [128]. *Asr* (*ABA*-stress response) is a family of genes involved in the *ABA*-dependent stress regulatory pathway. It was shown that the extent of nucleotide diversity in *Asr1* and *Asr2* genes in wild and cultivable common beans has depicted the adaptive selection of crops to cope with drought stress at varying levels [129].

In our study, *CDPK* was identified as one of the major hub-genes with 36 interactions, connecting with 27 protein families. During high-salt and extreme temperature conditions, calcium concentration in the cytosol was rapidly changed [130]. *CDPK* (Ser/Thr protein kinases) plays an important role in relaying the calcium signatures into downstream effects. Over-expression of *CDPK1* conferred tolerance to salinity and drought stress as reflected by the high percentage of seed germination, higher relative-water content, expression of stress-responsive genes, higher leaf chlorophyll content, increased photosynthetic efficiency, and other photosynthetic parameters in *Nicotiana tabacum* [130].

Several studies reported that the *NCED3* gene is a central enzyme in ABA biosynthesis. We analyzed *NCED*, which was found in connection with 40 other proteins in the PPI network. These proteins belonged to different protein kinases, detoxification factors, hormone signaling, and TFs. Over-expression of *NCED3* in Arabidopsis reduced the transpiration rate through increased ABA levels, which led to improved drought tolerance [131]. The involvement of *NCED* in drought response was found in avocado [132], common bean [92], tomato (*Solanum lycopersicum*) [133], and turmeric (*Curcuma longa*) [134]. The network showed a maximum number of intergroup interactions between molecular adaptation genes. The grouping of genes suggested that they could be involved in similar functions under drought stress in pearl millet.

The molecular physiology of drought tolerance in pearl millet can be perceived in two ways: genes that can provide overall plasticity to the plant system and genes that can work in specific traits and stages to impart tolerance. More experiments are needed to understand which one or combinations of perceptions would work under different production ecologies without comprising the grain and fodder yield. Another notable point is that heat stress is often accompanied by drought stress during critical stages such as seedling emergence, flowering, and post-flowering. It is also evident from several studies that some of the stress-tolerant pathways are common to both drought and heat stresses while other pathways are independent of each other. Identification of common traits or genes across stresses would help in developing robust genotypes. For example, leaf hairiness reduces leaf surface temperature as well as transpiration rate. Moreover, hairiness increases light reflectance, which minimizes the water loss under high temperatures. It also reduces radiation stress by increasing boundary layer resistance to water vapor movement away from the leaf surface. Other traits such as a better root system to mine water from shallow and deep soil, high-chlorophyll content to delay the senescence, and physiological traits such as potential evapotranspiration [135] to save water, among other effects, can be considered for developing better ideotypes for drought-prone ecologies.

It is also observed that plant growth and performance at later growth stages are positively associated with better seedling growth and its tolerance to drought stress. In most cases, seedling level tolerance could be translated to adult plant tolerance. The selection of seedlings having good tolerance to water stress is important while looking for tolerance at later stages since most of the functional mechanisms are in-built and will be constitutively expressed throughout the growth stages of the crop. Considering these assumptions, we have structurally and functionally characterized drought-responsive genes in pearl millet and other millet crops. The validated genes that we have identified can be used in a gene-editing experiment to augment the value of specific trait values under drought. Gene-specific SNPs can be identified from the genes, which will serve as important genomic resources in marker-assisted back-crossing and recurrent selection approaches [136] to develop new drought-tolerant lines. Through donors from wild species or landraces, drought tolerance can be introduced into cultivated elite lines [137,138]. Donor parents having the drought-tolerant SNPs can be used as founder lines in genomic prediction schemes [139] to develop drought-tolerant lines. Through the genomic selection, high-value SNPs will be accumulated in the lines and the superior genotypes with better breeding value will be used for the development of drought-tolerant varieties.

On the other hand, abiotic stresses, especially drought tolerance, is a complex quantitative trait where it is governed by several genes operating in different pathways. It is imperative to know that additional experiments are needed to mine stage-specific and trait-specific genes and SNPs to develop robust ideotypes that can produce higher grain yield across plant growth stages and ecologies under drought and combination abiotic stresses.

## Figures and Tables

**Figure 1 ijms-23-06907-f001:**
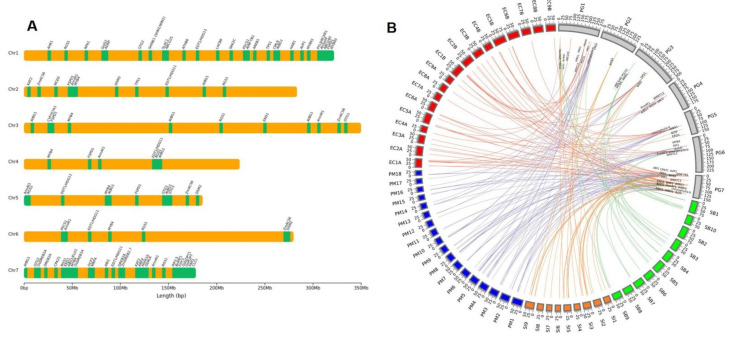
(**A**) Chromosomal distribution of 74 genes associated with drought in pearl millet. (**B**) Circos plot representing chromosomal positions of orthologous drought genes in different millet Crops.

**Figure 2 ijms-23-06907-f002:**
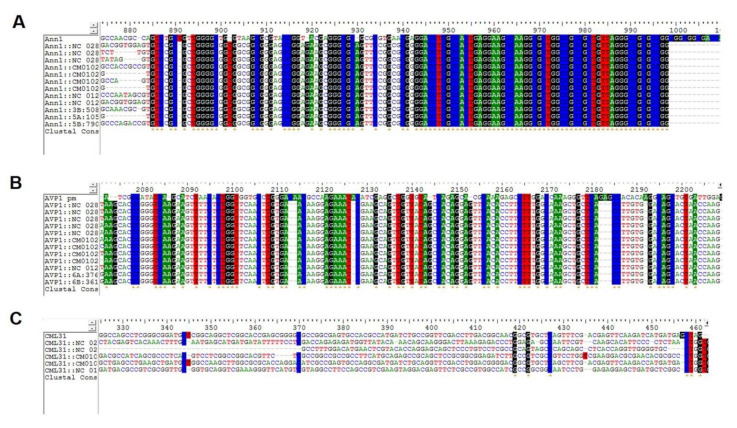
Haplotype-led comparison to visualize the occurrence of many to few haplotypes of drought genes in different millets. (**A**) Ann1 showed dense haplotype pattern (**B**) Moderate and scattered haplotypes in AVP1 (**C**) CML31 has the lowest number of haplotypes.

**Figure 3 ijms-23-06907-f003:**
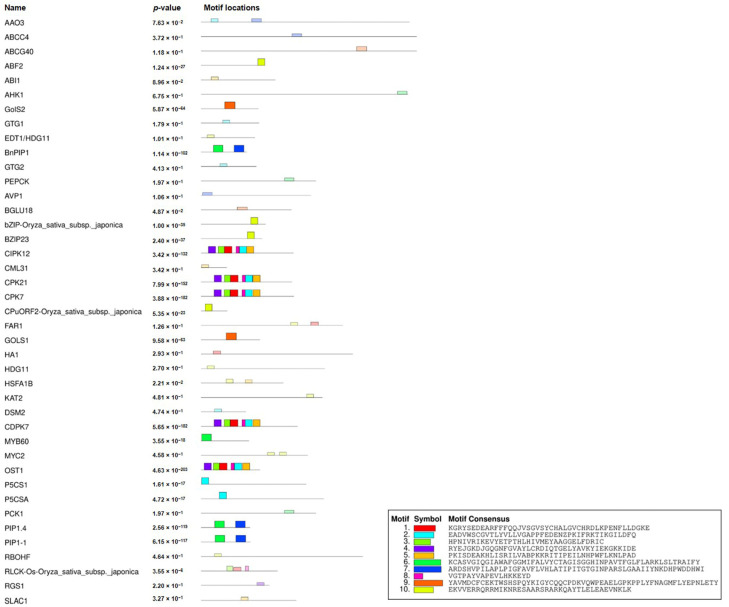
Presence of the conserved motifs in drought-related proteins. The consensus motif sequences are shown at the bottom.

**Figure 4 ijms-23-06907-f004:**
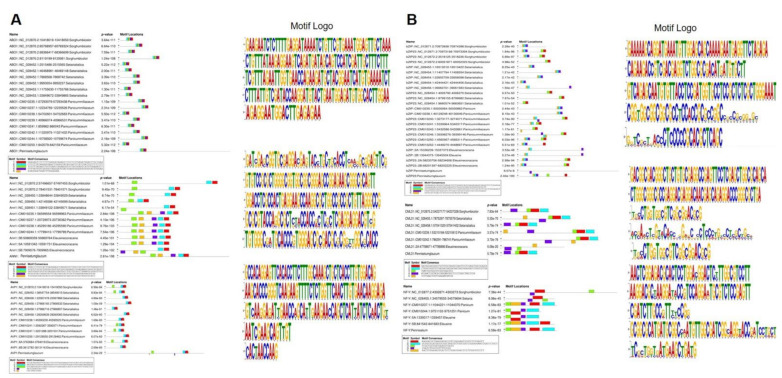
(**A**) Distribution and sequence of common conserved motifs of drought gene in pearl millet, sorghum, foxtail millet, proso millet, and finger millet. Clusters of motifs observed in similar order in different millets. Different colored boxes indicate different motifs. (**B**) Relatively less-conserved clusters of motifs observed in CML, Nf-Y, and bZIP genes among millet crops.

**Figure 5 ijms-23-06907-f005:**
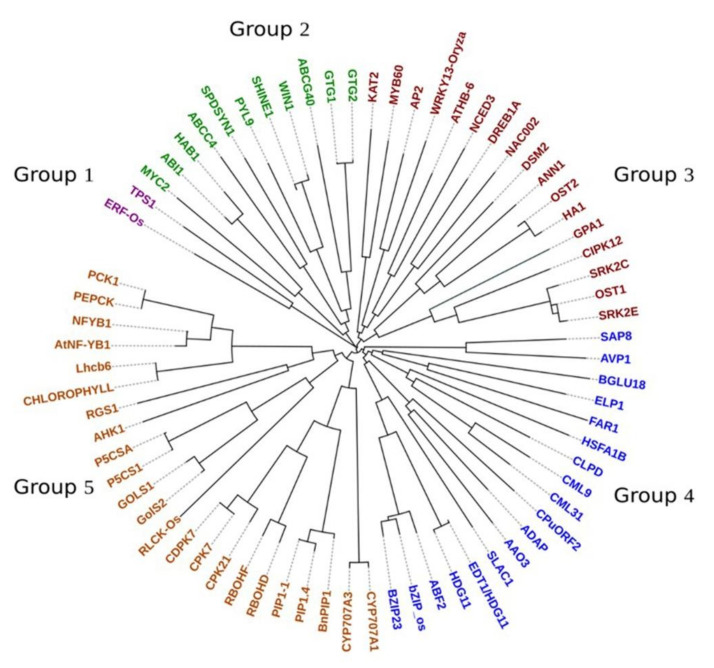
The phylogenetic tree of 74 drought genes in pearl millet grouped into five different clads.

**Figure 6 ijms-23-06907-f006:**
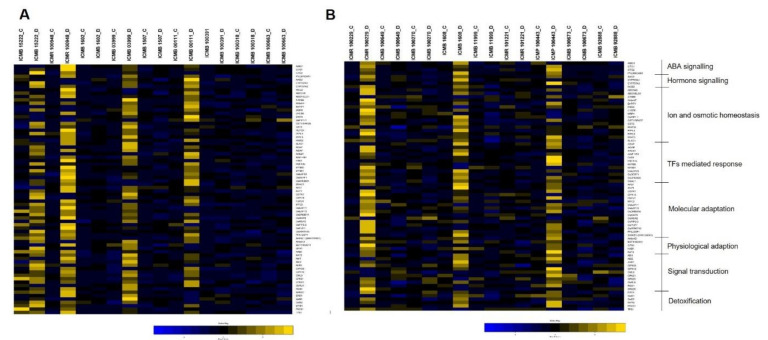
The heat map of 74 drought genes expressed in the (**A**) very-early and (**B**) early maturity group. The genotypes under drought (with suffix D) are compared with their respective control (with suffix C).

**Figure 7 ijms-23-06907-f007:**
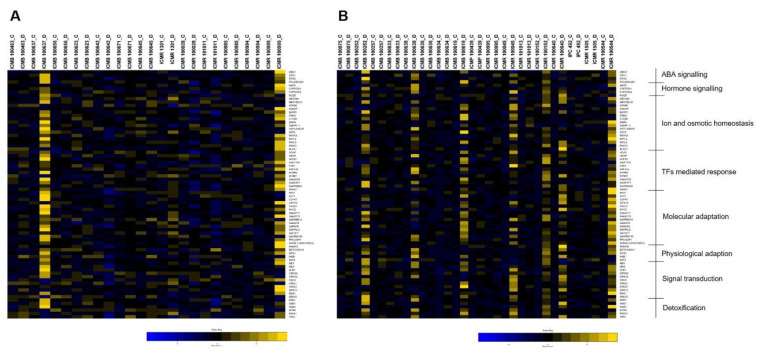
The heat map of 74 drought genes expressed in the (**A**) medium and (**B**) late maturity group. The genotypes under drought (with suffix D) are compared with their respective control (with suffix C).

**Figure 8 ijms-23-06907-f008:**
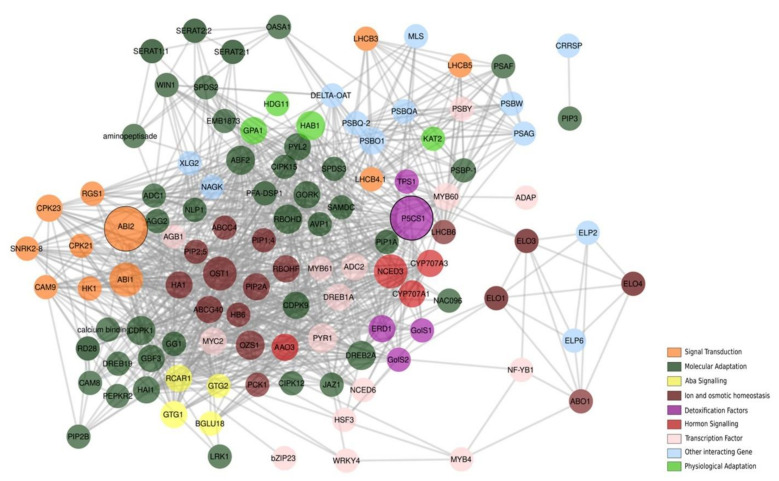
Protein–protein interaction network for 74 drought gene-based proteins. Color of the nodes indicates the functional class of the proteins.

**Table 1 ijms-23-06907-t001:** Important drought genes and their characteristics under various functional groups.

Group	Gene	Genotype	Maturity	Treatment		Function
				Control	Drought	
ABA signaling	*GTG2*	ICMP 100443	E	3.46	88.67	GPCR-type G protein; ABA-induced stomatal closure
	*AtBG1*	ICMB 100252	L	−6	88.05	beta-glucosidase; high level of ABA accumulation; stomatal movement
	*PYL9/RCAR1*	ICMB 100619	L	−4.31	84.63	ABA receptor; drought tolerance through reduced transpirational water loss and by inducing summer dormancy-like responses
Hormone signaling	*CYP707A3*	ICMB 00111	VE	−3.2	88.74	Involvement in ABA level regulation; stomatal responses, transpiration control
	*NCED*	ICMR 100948	VE	2.34	84.49	ABA biosynthesis enzyme; overexpression results in increased accumulation of ABA; reduced transpiration rate
Ion and osmotic homeostasis	*ATHB6*	ICMB 03999	VE	2.74	84.08	Involvement in seedling development during drought and osmotic stress.
	*OST1/SRK2E*	ICMB 15222	VE	−6.77	81.34	Kinase-like (open stomata 1), ABA-mediated stomatal aperture control, inhibition of ROS production
	*OST2*	ICMB 100619	L	4.56	88.05	Plasma membrane proton ATPase; involved in ABA-dependent pathway controlling stomatal closure
	*ABO1/ELO1*	ICMB 100252	L	−10.01	86.77	Multifunctional complex with roles in transcription elongation; Mediates ABA-induced stomatal closure
	*PIP1;4*	ICMR 100544	L	2.77	85.88	Plasma membrane intrinsic proteins; over-expression under drought causing reduced transpiration rate in leaves of *P. vulgaris* plant
TF-mediated	*ADAP*	ICMB 03999	VE	−6.08	88.3	Positive regulator of ABA; Regulation of seedling growth during water stress
	*HSFA1b*	ICMB 00111	VE	11.91	88.09	Heat shock proteins; protection of cellular proteins under drought stress
	*AREB1*	ICMR 100229	E	−2.89	87.36	Modulates endogenous ABA level and ROS level in transgenic Arabidopsis
	*MYB60*	ICMB 100638	L	−10.6	89.57	Drought tolerance via an increase in lateral root growth facilitating more water uptake, controlling stomatal movement
	*FAR1*	ICMR 100045	L	−3.43	89.56	Over-expression under drought; ROS scavenging
	*SNAC1*	ICMR 100544	L	−3.01	89.02	Accumulation in guard cells; Over-expression reduced transpirational losses due to increased stomatal closure
	*AlSAP*	ICMB 100638	L	−5.93	85.75	Leaf rolling, improved tillering, grain yield under drought resulting in accumulation of green biomass during vegetative growth
Molecular adaptation	*OsWRKY45*	ICMR 100229	E	2.26	87.8	Induced by ABA; positively regulated under drought; involvement in signaling pathway; stomatal closure
	*FSPD1*	ICMP 100443	E	−2.98	84.98	Spermidine synthetase gene; overexpression of spermine in roots and leaves resulting in drought tolerance
	*OsPIP2-2*	ICMB 100619	L	2.6	85.2	Aquaporin; osmoprotectant involved in drought tolerance
Physiological adaptation	*KAT2*	ICMP 100443	E	2.52	88.54	KAT2 over-expressing transgenic lines showed ABA-induced stomatal closure; inhibition of stomatal opening
	*HAB1*	ICMB 100638	L	−16.08	84.05	Accumulation of ABA under stress; stomatal closure to prevent water loss
	*EDT1/HDG11*	ICMB 100638	L	−5.94	87.8	Enhanced tolerance to drought in transgenic cotton and poplar; improved root system, accumulation of proline, soluble sugars, antioxidant enzymes in cotton, and stomatal traits.
Signal transduction	*RGS1*	ICMR 100999	M	9.74	86.87	Increase in root growth, regulation of stomatal traits such as closure of stomata, low stomatal density, and small stomatal aperture cause tolerance to drought.
	*SRK2C*	ICMB 100252	L	−6.74	84.53	Osmotic stress-activated protein kinase; controlling stomatal aperture
Detoxification	*TPS1*	ICMR 100948	VE	−5.14	87.58	Over-expression under drought; causing accumulation of trehalose, decreased stomatal density, and reduced transpiration rate in maize
	*ERD1*	ICMB 100252	L	−3.04	82.26	Chloroplast-targeted Clp protease; functions with NAC and ZFHD1 to improve drought tolerance.
	*GolS2*	ICMR 100544	L	2.07	78.98	Drought tolerance via accumulation of galactinol and raffinose in crops, causing a reduced rate of transpiration from leaves.
	*P5CS1*	ICMB 100619	L	−3.15	75.64	ABA-mediated expression; role in the drought-induced accumulation of proline

## Data Availability

Data is contained within the article or Appendix A.

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
