# Peer review of "Identification of Candidate Genes Regulating Drought Tolerance in Pearl Millet"

_ijms, 2022, doi:10.3390/ijms23136907_

Round 1

Reviewer 1 Report

Authors have considerably improved the quality of the manuscript and discussed all my previous concerns. However, authors should improve the quality/resolution of the figures because they are mostly illegible or too small to be clearly readable.  

Author Response

We sincerely thank the Reviewer for the constructive and well-thought-out comments and suggestions to improve the manuscript.

We shall be able to provide high-resolution images if it is needed.

Reviewer 2 Report

The authors studied the identification of candidate genes regulating drought tolerance in 48 inbreds pearl millet at the structural and functional levels. The authors showed great work.

I have found some comments in the manuscript that need to be addressed and clarified:

-The abstract must be supported with more results

-  Drought treatment was very late.

- On what basis was the list of candidate drought-related genes elaborated?

- The gene organization of the predicted drought-related genes was absent and must be developed. 

-The transcript accumulation level of (74) drought-related genes were conducted for two groups under normal and drought stress conditions. Perhaps now, with the reduced cost of NGS, an RNAseq study will be more appropriate for an overview of the genes regulation under drought stress conditions and facilitates the discovery of drought-related genes

Author Response

We sincerely thank the Reviewer for the constructive and well-thought-out comments and suggestions to improve the manuscript. We have agreed all the suggestions and revised the manuscript accordingly.

Reviewer 2:

The authors studied the identification of candidate genes regulating drought tolerance in 48 inbreds pearl millet at the structural and functional levels. The authors showed great work.

I have found some comments in the manuscript that need to be addressed and clarified:

-The abstract must be supported with more results

Ans: Agreed, we have added additional details in the abstract.

-  Drought treatment was very late.

Ans: In this experiment, we wish to target one of the important key phenological stages of pearl millet production ecology, which is flowering, and is very crucial to sustain the fertile florets and grain yield. During drought, this critical stage is even more significant so we have conducted the experiment in lysi-meter and imposed drought during flowering stage in order to capture the genes expressed in that specific stage. The responsive genes identified during flowering will be used in breeding programs to develop tolerant varieties.

- On what basis was the list of candidate drought-related genes elaborated?

Ans: Identification of homologous genes from the public domain and exploring their utility in other species is a well-known process to understand the genetics of traits. Pearl millet has been sequenced during 2017 and hardly a couple of articles on gene function and gene regulation at the whole genome level were published. Considering the limited functional genomic resources, searching genes from the public domain is inevitable to understand trait genetics. In this context, we have short-listed 171 genes for the experiment from the public domain which were well characterized and extensively studied at different physiological and molecular adaptation levels in different crops.  Relying on the maximum homology identified between the annotated genes and pearl millet, a set of 74 genes were selected belonging to different functional classes.

- The gene organization of the predicted drought-related genes was absent and must be developed.

Ans: Agreed. We have added the complete gene organisation of the genes and presented in Supplementary Table 3 considering the size of the table.

 -The transcript accumulation level of (74) drought-related genes were conducted for two groups under normal and drought stress conditions. Perhaps now, with the reduced cost of NGS, an RNAseq study will be more appropriate for an overview of the genes regulation under drought stress conditions and facilitates the discovery of drought-related genes  

Ans: The whole-genome transcriptome experiments such as RNASeq usually end up in thousands of genes regulating different functions using a couple of genotypes. Those kinds of experiments finally report a small set of genes, probably 5 to 10%, related to the specific traits for which the experiment is meant for. Even after conducting a genome-wide transcriptome experiment, Reviewers usually suggest to validate the genes in two ways: 1. through other genomic experiments such as QTL mapping GWAS/etc and 2. validate a set of potential genes identified by RNASeq through RT-PCR experiments. In fact, RT-PCR is the gold standard for expression analysis and will be used to confirm results from HTP gene expression experiments.

The objective of the experiment is not to search novel genes, while, our experiment is very specific with focused objectives to validate the drought-responsive genes in 48 genotypes having a variable response to drought stress. Our experiment provided meaningful information and functional characterization of drought-tolerant genes in pearl millet and will be directly helpful in drought breeding programs.

Reviewer 3 Report

The work by Chakraborty et al. advances our understating of water use efficiency and its genetic transcriptomic basis in pearl millet for drought tolerance at a key phenological stage such as flowering. The report is well written and condensed, as well as technically appropriate.  However, before being able to recommend acceptance, I invite authors to address the following amendments.

First, the introduction section should properly close (P5) with an explicit research questions and hypotheses. I just would recommend adding a short sentence before to emphasize the research gap that inspired pursuing this work pearl millet.

Second, materials and methods must start (second section in P6) with a short sentence commenting on a preliminary statistical power analysis, specially taking into account that the strong influence of genotypic sampling on RNA-based studies. How representative are they? Please explicitly re-visit this sampling caveat within the discussion section (P39). The report is so far lacking a very brief closing paragraph (also in P39) that describes the major caveats/limitations of the present work. It is never beyond the scope of any research to explicitly acknowledge the study caveats, especially when dealing with highly variable RNA expression profiles. Also link to this point, a short perspectives section, just before the conclusions (in P39), would be insightful for readers to fill the identified caveats as part of future research. Specifically, what other accessions, traits and combined abiotic stresses must future studies target in pearl millet? How could the identified candidate genes source pearl millet improvement programs?

Third, concerning the results, figures are very well edited and insightful. Still, authors should improve Figures 5 so that the tree depicts bootstrap values on the branches. This would improve its confidence.

Fourth, the discussion, although perceptive, should embrace broader reflections on whether drought tolerance in pearl millet could be understood as a plastic or alternative a genetic adaptive strategy to cope with water scarcity.

Besides, since abiotic stresses are usually pleiotropic, please also comment on potential gene functional correlates with heat tolerance, which is a stress typically correlated with drought in the face of changing climate (refer to PLoS One 2013 8(5):e62898). What about combined drought + heat treatments to study the same pearl millet panel? Also related to this point on pleiotropy, authors must also briefly mention whether adaptive trade-offs for drought tolerance are observed/predicted (check and comment in the light of Front Plant Sci 2018 9:128). For instance, a biomass/reproductive trade-off in pearl millet may also be evident in more subtle ways such as plant architecture and nutrient accumulation in the face of drought stress. For instance, plants’ architecture is known to prevent soil desiccation.

Last but not least, although the paper provides good evidence into the molecular mechanisms for drought tolerance in pearl millet, a major question that authors should prospect in their discussion is how pearl millet improvement for drought tolerance may unlock and effectively utilize these mechanisms (see as guidance Genes 2021 12:556). Gene editing, recurrent backcrossing and inter-specific schemes (condensed, contrasted and discussed in Front Genet 2020 11:564515, and Genes 2021 12:783, please include both) may offer an avenue in this regard that authors should acknowledge (refer to Agronomy 2021 11:1978). Please envision any other recommendation.

Some additional minor comments are enlisted below:

- P5, first paragraph; P28, first paragraph; P35, first paragraph. When mentioning/discussing heat shock proteins (HSP), please also comment in the light of the work Front Genet 2019 10:954, which also found the same genetic architecture.

- P5, second paragraph; P15, last paragraph; P10, second paragraph; P29, first and last paragraphs; P36, second paragraph; P37, second and third paragraphs; P38, first paragraph; among others. When discussing DREB-based ABA-independent drought tolerance responses, please also refer to the work Theor Applied Genet 2012 125(5):1069-85.

- P5, second paragraph. Why is the drought during the flowering phenological phase focus particularly insightful? What about terminal drought? Please expand, especially for reader coming more from a breeding side, non-specialized in physiology. 

- P38, second paragraph. Please comment/discuss in the light of other ABA-mediated drought tolerance pathways such as ASR genes (refer to BMC Genet 2012 13:58).

- What can authors tell about ERECTA-mediated drought tolerance? This is also another common drought tolerance pathway across several crops (i.e., refer to Plant Sci 2016 242: 250). Any insight in pearl millet?

- As a last point, in order to make revision of the manuscript easier, please make sure to include line numbers in the next version.

Author Response

We sincerely thank the Reviewer for the constructive and well-thought-out comments and suggestions to improve the manuscript. We have agreed all the suggestions and revised the manuscript accordingly.

Reviewer 3:

The work by Chakraborty et al. advances our understating of water use efficiency and its genetic transcriptomic basis in pearl millet for drought tolerance at a key phenological stage such as flowering. The report is well written and condensed, as well as technically appropriate.  However, before being able to recommend acceptance, I invite authors to address the following amendments.

First, the introduction section should properly close (P5) with an explicit research questions and hypotheses. I just would recommend adding a short sentence before to emphasize the research gap that inspired pursuing this work pearl millet.

Ans: Agreed and we have refined the draft.

Second, materials and methods must start (second section in P6) with a short sentence commenting on a preliminary statistical power analysis, specially taking into account that the strong influence of genotypic sampling on RNA-based studies. How representative are they? Please explicitly re-visit this sampling caveat within the discussion section (P39). The report is so far lacking a very brief closing paragraph (also in P39) that describes the major caveats/limitations of the present work. It is never beyond the scope of any research to explicitly acknowledge the study caveats, especially when dealing with highly variable RNA expression profiles. Also link to this point, a short perspectives section, just before the conclusions (in P39), would be insightful for readers to fill the identified caveats as part of future research. Specifically, what other accessions, traits and combined abiotic stresses must future studies target in pearl millet? How could the identified candidate genes source pearl millet improvement programs?

Ans: Agreed and we have made changes in the draft.

Third, concerning the results, figures are very well edited and insightful. Still, authors should improve Figures 5 so that the tree depicts bootstrap values on the branches. This would improve its confidence.

Ans: Agreed. We have added boot-strap values on the branches.

Fourth, the discussion, although perceptive, should embrace broader reflections on whether drought tolerance in pearl millet could be understood as a plastic or alternative a genetic adaptive strategy to cope with water scarcity.

Besides, since abiotic stresses are usually pleiotropic, please also comment on potential gene functional correlates with heat tolerance, which is a stress typically correlated with drought in the face of changing climate (refer to PLoS One 2013 8(5):e62898). What about combined drought + heat treatments to study the same pearl millet panel? Also related to this point on pleiotropy, authors must also briefly mention whether adaptive trade-offs for drought tolerance are observed/predicted (check and comment in the light of Front Plant Sci 2018 9:128). For instance, a biomass/reproductive trade-off in pearl millet may also be evident in more subtle ways such as plant architecture and nutrient accumulation in the face of drought stress. For instance, plants’ architecture is known to prevent soil desiccation.

Last but not least, although the paper provides good evidence into the molecular mechanisms for drought tolerance in pearl millet, a major question that authors should prospect in their discussion is how pearl millet improvement for drought tolerance may unlock and effectively utilize these mechanisms (see as guidance Genes 2021 12:556). Gene editing, recurrent backcrossing and inter-specific schemes (condensed, contrasted and discussed in Front Genet 2020 11:564515, and Genes 2021 12:783, please include both) may offer an avenue in this regard that authors should acknowledge (refer to Agronomy 2021 11:1978). Please envision any other recommendation.

Ans: We have added additional details based on the above-mentioned line of thoughts.

Some additional minor comments are enlisted below:

- P5, first paragraph; P28, first paragraph; P35, first paragraph. When mentioning/discussing heat shock proteins (HSP), please also comment in the light of the work Front Genet 2019 10:954, which also found the same genetic architecture.

Ans: Agreed and discussed in the draft.

- P5, second paragraph; P15, last paragraph; P10, second paragraph; P29, first and last paragraphs; P36, second paragraph; P37, second and third paragraphs; P38, first paragraph; among others. When discussing DREB-based ABA-independent drought tolerance responses, please also refer to the work Theor Applied Genet 2012 125(5):1069-85.

Ans: Agreed and discussed in the draft.

- P5, second paragraph. Why is the drought during the flowering phenological phase focus particularly insightful? What about terminal drought? Please expand, especially for reader coming more from a breeding side, non-specialized in physiology. 

Ans: Agreed and discussed in the draft.

- P38, second paragraph. Please comment/discuss in the light of other ABA-mediated drought tolerance pathways such as ASR genes (refer to BMC Genet 2012 13:58).

Ans: Agreed and discussed in the draft.

- What can authors tell about ERECTA-mediated drought tolerance? This is also another common drought tolerance pathway across several crops (i.e., refer to Plant Sci 2016 242: 250). Any insight in pearl millet?

Ans: The ERECTA genes would be of interesting candidates to control stomatal density and related transpiration rate thereby achieving drought tolerance. When we have searched in literature, we couldn’t find ERECTA-mediated drought genes in pearl millet. We wish to focus on more of such genes in pearl millet in upcoming experiments.

- As a last point, in order to make revision of the manuscript easier, please make sure to include line numbers in the next version.

Ans: Agreed, we have included line numbers in the revised draft.

Round 2

Reviewer 3 Report

Thanks for following the recommendations and preparing this updated version. New figure 5 with bootstrapsupport is ideal. At production, please make sure figure resolution is high to guarantee readability